# Parameter calibration of the discrete element simulation model for soaking paddy loam soil based on the slump test

Zhou Tienan [1], Hao Zhou[1,2]*, Jiangtao Ji[1,2], Fengyun Sun[1], Zhiyu Qin[1]

1 College of Agricultural Equipment Engineering, Henan University of Science and Technology, Luoyang, China, 2 Collaborative Innovation Center of Machinery Equipment Advanced Manufacturing of Henan Province, Luoyang, China

* zhhao_2018@163.com

**Data Availability Statement:** All relevant data are within the manuscript and its Supporting Information files.

## Abstract

The discrete element computer simulation method is an effective tool that enables the study of the interaction mechanism between the pulping components and the paddy soil during the paddy field pulping process. The findings are valuable in optimizing the parameters of the paddy beating device to improve its working quality and efficiency. However, the lack of accurate soil models for paddy soil has limited the application and development of the discrete element method in paddy pulping research. This study selected the Hertz-Mindlin with Johnson-Kendall-Roberts discrete element model for the pre-pulping paddy loam soil and used the slump error as the test index to select nine parameters, including soil Poisson's ratio and surface energy, as test factors to calibrate the model parameters. The Plackett-Burman test identified soil shear modulus, surface energy, and soil-iron plate static friction coefficient as significant factors affecting the test index. The steepest ascent test results determined the test range of the above parameters. The Box-Behnken test obtained the regression model between the significant factors and the test index, and the regression model was optimized using the slump error as the target. The optimal combination of parameters was surface energy of 3.257 J/m², soil shear modulus of 0.709 MPa, and static friction coefficient between soil and iron plate of 0.701. The slump simulation test using this combination of parameters yielded an average slump error of 2.04%. The collective results indicate the accuracy of the calibrated discrete element simulation parameters for paddy loam soil. These parameters can be used for discrete element simulation analysis of the paddy pulping process after paddy field soaking.

## 1 Introduction

Rice is an important food crop in China. Cultivation of rice requires paddy field pulping to improve the quality of paddy soil and support rice growth [1, 2]. However, problems with the existing paddy beater include low operational efficiency and poor operational quality in the pulping process, which seriously affects the efficiency and quality of rice cultivation [3].

**Funding:** This study was supported by the National Natural Science Foundation of China (Grant No. 52005162). The funding organization had no involvement in the study design, data collection and analysis, decision to publish, or preparation of the manuscript.

**Competing interests:** The authors have declared that no competing interests exist.

Therefore, the development of high-performance paddy field pulping machines is crucial for high-quality and efficient rice cultivation.

To develop these machines, the interaction mechanism between the machinery and the soil must be studied to optimize the structural and working parameters of the pulping parts [4]. The discrete element method allows the study and analysis of the interaction mechanism between the pulping part and the soil of the paddy field. The data inform the optimization of the parameters of the pulping part [5]. Jiang et al. used discrete element modeling software (EDEM™) software to construct a straw-aqueous layer-mud-soil model, and studied the difference between the working parameters of the pulping mechanism on the forward resistance and the initial pulping concentration to obtain better working parameters. The authors verified that the error between the actual test and simulation results was 11.64% [6]. Chen et al. performed a simulation analysis of the throwing behavior of the pulping blade in response to the current situation of poor side throwing effect of the pulping blade in the paddy field. The findings verified that the maximum error of throwing quality and throwing distance was 9.48% and 13.10%, respectively [7]. The above studies investigated the influence of working parameters on the resistance and pulping concentration, and sludge-throwing behavior of the pulping blade, and proposed theoretical research directions for optimizing the design. However, there are cases of large errors between the test and simulated values (error values >9%). To adapt the pulping simulation study and improve the accuracy of the simulation results, discrete element modeling and parameter calibration of the soil in the paddy field after soaking is required [8]. Li et al. analyzed the static friction coefficient of viscous heavy black soils with different water contents to illustrate the effect of water content on soil adhesion and calibrated the black soil parameters [9]. Xing et al. calibrated the key parameters of the discrete element model for the cohesive Hainan brick red clay soil and validated the model by the fracture resistance test with a test error of 3.43% [10]. Tian et al. calibrated the coefficient of friction between materials, the coefficient of friction between materials and steel, and the surface energy by targeting the resting angle of a maize straw-soil mixture pile in a black soil area [11]. Wang et al. used an elastoplastic contact model to calibrate the no-till clay soil in the wheat and maturing zone of North China and determined the best combination of values for particle radius, interparticle rolling friction factor, and static friction factor, which was verified by comparative simulation tests [12]. Shi et al. integrated the delayed elasticity model with the linear cohesion model to establish a model for agricultural soils in the northwest arid region and predicted the shear strength, interparticle static friction factor, and dynamic friction factor by water content [13]. Ding et al. used the Hertz-Mindlin with bonding contact model to establish a soil discrete element model applicable to deep loosening tillage of cohesive rice soils. The authors also used a simulation study to analyze the mechanism of deep loosening of paddy soils, and compared and verified the tillage resistance of implements and soil disturbance with the results of field tests [14]. The above studies were intended to calibrate and validate the parameters of discrete element models for different soil types. These studies were able to simulate the mechanical behavior of soils more accurately by calibrating parameters, such as the friction coefficient between soil particles and the material surface energy. However, little research has been performed to calibrate discrete element models for paddy loam soils with high water content.

The lack of accurate discrete element models for paddy loam soils prompted us to use a discrete element simulation model calibrated for paddy loam soils before pulping to provide a more accurate discrete element simulation base model capable of studying the interaction mechanism between pulping parts and paddy loam soil during the pulping process. In this paper, the development of a discrete element model for post-soaking paddy loam soils is described. Based on the slump test, the slump error was used as the test index. Nine

parameters, including Poisson's ratio, shear modulus, and surface energy of the soil, were used as the test factors. The Plackett-Burman, steepest ascent, and Box-Behnken tests were performed in turn to calibrate the discrete element simulation parameters for post-soaked paddy loam soils. The findings allowed the calibration of the discrete elements of soil after soaking.

## 2 Materials and methods

### 2.1 Soil parameter determinations

**2.1.1 Determination of basic physical parameters.** The clay loam soil sample is a typical soil type in the rice-wheat rotation area. The discrete element parameter calibration of the soil can help the application and development of the discrete element method in puddling research [15]. In the present study, the soil was randomly sampled after soaking for 24 h. The mean water content was 32%. The soil density was measured using a $1×10^{-4}$ m$^3$ ring knife and an electronic balance with a range of 500 g and accuracy of 0.01 g. The mean value was 1808 kg/m$^3$.

**2.1.2 Soil slump test.** A pre-test established that the soaking paddy loam soil has characteristics of high cohesiveness and fluidity. Therefore, in this paper, the slump test was used as the basic test for paddy loam soil calibration. The slump of the paddy loam soil was used to determine the characteristics of the paddy loam soil after soaking. The slump error of the paddy loam soil was used as the test index to calibrate the paddy loam soil simulation parameters [16]. The slump of the paddy loam soil was measured using a slump cone. A slump cone 300 mm in height, with top and bottom diameter of 100 and 200 mm, respectively, was placed on a leveled base plate. The paddy loam soil was loaded into the slump cone and pounded to compact it. The excess soil was scraped off the top. The slump cone was then lifted at a constant speed within 5s. The slump was measured when the soil finished slumping. The determined value represented the height of the slump cone minus the reading of the tape measure (Fig 1); the average value of five repeat determinations was recorded; the average value of soil slump in the paddy field was 177.4 mm.

Construct a three-dimensional model in a 1:1 scale based on the actual dimensions of the slump cone and import it into the EDEM discrete element simulation software. Simulation of the paddy loam soil was performed according to the slump test process. After the paddy loam

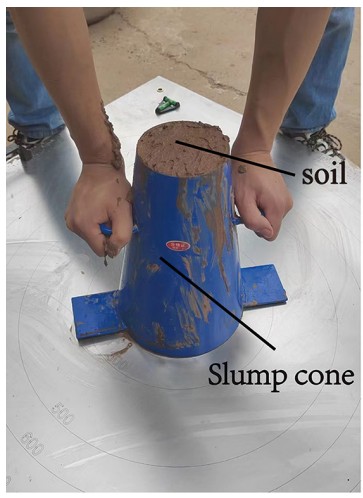
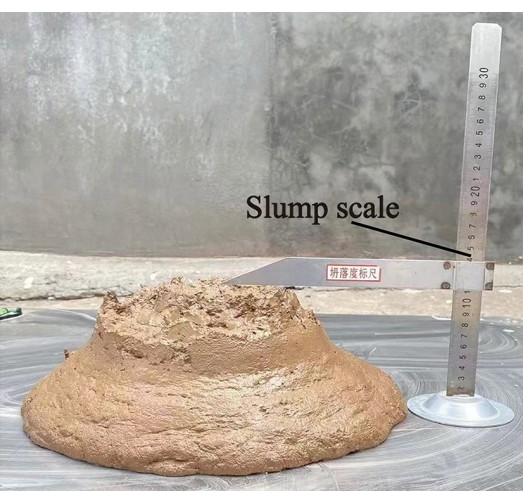
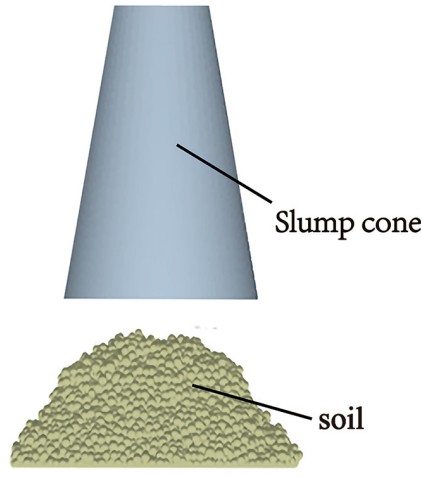

a. Slump physical test b. Slump measurement c. Simulation of slump test

**Fig 1. Slump test of paddy loam soil.**

soil stopped flowing, the slump was measured using the post-processing function of EDEM. The simulation of the slump test is shown in Fig 1c.

## 2.2 Discrete element simulation model and calibration of parameters

**2.2.1** Discrete element simulation model. Hertz-Mindlin with Johnson-Kendall-Roberts (JKR) is a commonly used model for analyzing cohesive wet soils. The analysis is based on the Hertz-Mindlin (no slip) contact model, which takes into account the effects of plastic deformation between particles and interparticle adhesion forces on particle motion patterns [17, 18]. Wu Tao et al. productively used this contact model in the simulation of cohesive soils [19]. Due to the highly cohesive nature of paddy loam soils after soaking, the Hertz-Mindlin with JKR model was presently chosen as the contact model for discrete element soil modeling.

In the JKR contact model, the normal elastic contact force between particles, $F_{JKR}$, is calculated as:

$$F_{JKR} = -4\sqrt{\pi\gamma E^*}\alpha^{\frac{3}{2}} + \frac{4E^*}{3R^*}\alpha^3 \tag{1}$$

The relationship between the radius $\alpha$ of contact between particles and the amount of overlap $\delta$ is calculated as:

$$\delta = \frac{\alpha^2}{R^*} - \sqrt{\frac{4\pi\gamma\alpha}{E^*}} \tag{2}$$

The equivalent contact radius $R^*$ and equivalent modulus of elasticity $E^*$ is calculated as:

$$\frac{1}{E^*} = \frac{1 - V_1^2}{E_1} + \frac{1 - V_2^2}{E_2} \tag{3}$$

$$\frac{1}{R^*} = \frac{1}{R_1} + \frac{1}{R_2} \tag{4}$$

where $F_{JKR}$ is the JKR normal contact force, N; $\gamma$ is the surface energy, J/m$^2$; $E^*$ is the equivalent modulus of elasticity; $R^*$ is the equivalent contact radius, mm; $V_1$ and $V_2$ are the particle Poisson's ratios; $R_1$ and $R_2$ are the particle radii in mm.

As the soil properties of the references were similar to those of the soils in this study, the range of values for the present paddy loam soils was obtained by integrating the parameter values of the soils calibrated in the course of the previous study. Based on the literature, the nine parameters to be calibrated and their range of values were as follows: Poisson's ratio 0.3~0.5, surface energy 0.01~3.5 J/m$^2$, shear modulus 0.1~1 Mpa, soil-soil recovery coefficient 0.01~0.5, soil-soil static friction coefficient 0.1~0.9, soil-soil rolling friction coefficient 0.01~0.5, soil-iron plate recovery coefficient 0.01~0.5, soil-iron plate static friction coefficient 0.1~0.9, and soil-iron plate rolling friction coefficient 0.01~0.5 [9, 10, 19–21]. Spherical particles with a diameter of 5 mm, particle density of 1808 kg/m$^3$, and particle size distribution of 0.8~1.2 were selected for the simulation. The material of the slump cone and the bottom plate was defined as steel with a density of 7850 kg/m$^3$, Poisson's ratio of 0.3, and shear modulus of 7×10$^{10}$ pa [22, 23].

**2.2.2. Simulation parameter calibration.** The slump error of paddy loam soil was used as a test index to calibrate the discrete element simulation parameters. The simulation parameters with significant influence on the slump error were screened out by the Plackett-Burman test. The steepest ascent test was then used to approximate the better value range of the significant parameters. Finally, the Box-Behnken test was used to obtain the regression model between

the test parameters and the test index. The regression equation was optimized to obtain the best parameter combination, and the combination was verified by simulation tests. The slump error is defined as:

$$\varphi = \frac{|Y_1 - Y_2|}{Y_2} \times 100\% \tag{5}$$

where $\varphi$ is the slump error, %; $Y_1$ and $Y_2$ is the simulated and measured values of soil slump in paddy fields, respectively, in mm.

## 3 Results

### 3.1 Plackett-Burman test

The Plackett-Burman test allows for the rapid screening of test factors that have a significant impact on an indicator under conditions of a larger number of test factors. In this test, nine simulation parameters were selected. Each parameter was taken at high and low levels (+1 and -1) according to the range of values, as shown in Table 1. The Plackett-Burman test design was performed using Design-Expert software according to the parameter levels in Table 1. The test scheme and results are shown in Table 2. Significance analysis of the test results was performed using Design-Expert. The significance of the effect of each parameter on the test indices is presented in Table 3.

As shown in Table 3, the contribution of the nine test factors to the test index was ranked from highest to lowest. The order was *C*, *B*, *H*, *I*, *G*, *E*, *A*, *D*, *F*. Among them, the p-values of *B*, *C*, and *H* were <0.05, indicating significant effects on the test index.

### 3.2 Steepest ascent test

The steepest ascent test allows quick access to the factor interval around the optimum value of the test indicator. The effective range of values for the three significant factors were investigated by screening using this test. The steepest ascent test protocol and results are shown in Table 4.

The slump error decreased with the increase of each factor level and reached the minimum value at level 4. Thereafter the slump error increased with increasing factor level; the simulation parameter range interval was better around level 4. Therefore, level 4 represented the center point, and levels 3 and 5 the parameter intervals for subsequent calibration tests. The remaining parameters represented the intermediate values of the parameters with less influence on the slump error as the subsequent simulation parameters [9, 10], Poisson's ratio of 0.4, soil-soil recovery coefficient of 0.25, soil-soil static friction coefficient of 0.5, soil-soil rolling

**Table 1. Factors and level of Plackett-Burman test.**

| Element | Parameters | Low level (-1) | High level (1) |
|---|---|---|---|
| *A* | Poisson's ratio of soil | 0.3 | 0.5 |
| *B* | Surface energy of soil (J/m$^2$ | 0.01 | 3.5 |
| *C* | Shear modulus of soil (Mpa) | 0.1 | 1 |
| *D* | Coefficients of soil–soil restitution | 0.01 | 0.5 |
| *E* | Coefficients of soil–soil static friction | 0.1 | 0.9 |
| *F* | Coefficients of soil–soil rolling friction | 0.01 | 0.5 |
| *G* | Coefficients of soil–steel restitution | 0.01 | 0.5 |
| *H* | Coefficients of soil–steel static friction | 0.1 | 0.9 |
| *I* | Coefficients of soil–steel rolling friction | 0.01 | 0.5 |

**Table 2. Scheme and result of Plackett-Burman test.**

| No. | a | b | c | d | e | f | g | h | i | Slump error /% |
|-----|-----|-----|-----|-----|-----|-----|-----|-----|-----|----------------|
| 1 | 1 | 1 | -1 | 1 | 1 | 1 | -1 | -1 | -1 | 30.58 |
| 2 | -1 | 1 | 1 | -1 | 1 | 1 | 1 | -1 | -1 | 14.41 |
| 3 | 1 | -1 | 1 | 1 | -1 | 1 | 1 | 1 | -1 | 16.17 |
| 4 | -1 | 1 | -1 | 1 | 1 | -1 | 1 | 1 | 1 | 20.95 |
| 5 | -1 | -1 | 1 | -1 | 1 | 1 | -1 | 1 | 1 | 24.31 |
| 6 | -1 | -1 | -1 | 1 | -1 | 1 | 1 | -1 | 1 | 35.1 |
| 7 | -1 | -1 | -1 | -1 | 1 | -1 | 1 | 1 | -1 | 29.44 |
| 8 | -1 | 1 | -1 | -1 | -1 | 1 | -1 | 1 | 1 | 24.95 |
| 9 | 1 | 1 | 1 | -1 | -1 | -1 | 1 | -1 | 1 | 17.65 |
| 10 | -1 | 1 | 1 | 1 | -1 | -1 | -1 | 1 | -1 | 4.62 |
| 11 | 1 | -1 | 1 | 1 | 1 | -1 | -1 | 1 | 1 | 30.67 |
| 12 | -1 | -1 | -1 | -1 | -1 | -1 | -1 | 1 | -1 | 36.88 |

friction coefficient of 0.25, soil-iron plate recovery coefficient of 0.25, and soil-iron plate rolling friction coefficient of 0.25.

## 3.3 Box-Behnken test

The Box-Behnken test is a response surface test design that effectively explores the effects of factors within a factor range on a test index, and the interactions between factors. The test determines the combination of factors that results in the optimal value of the test index. The slump error of paddy loam soil was used as the test index. The Box-Behnken test was conducted with surface energy, soil shear modulus, and soil-soil static friction coefficient as the test factors. The test factor coding table is shown in Table 5. The Box-Behnken test design was performed using Design-Expert software according to the parameter levels in Table 5. The test scheme and results are shown in Table 6. Multiple regression analysis was performed on the test results using Design-Expert software to obtain the slump error regression model, which was subjected to analysis of variance. The results are shown in Table 7.

The data in Table 7 indicate that the effects of $B$, $H$, $BC$, $BH$, $CH$, $B^2$, $C^2$, and $H^2$ on slump error are highly significant (P < 0.01), while the effect of $C$ on slump error is not significant (P > 0.05). The slump error fitted regression model and misfit term P-value of < 0.01 and > 0.05, respectively, indicate that the model fits well with no misfit phenomenon. The coefficient of determination $R^2$ of the regression equation was 0.9915, and the corrected coefficient of determination adj-$R^2$ was 0.9805. These findings indicate that the regression model

**Table 3. Significance analysis of parameters in the Plackett-Burman test.**

| Element | Effect | Contribution/% | Sum of Squares | F | P | Significance Rank |
|---------|--------|----------------|----------------|------|------|-------------------|
| A | 2.20 | 1.47 | 14.52 | 5.85 | 0.136714 | 7 |
| B | -9.90 | 29.84 | 294.16 | 118.57 | 0.008329** | 2 |
| C | -11.68 | 41.52 | 409.33 | 164.98 | 0.006007** | 1 |
| D | -1.59 | 0.77 | 7.60 | 3.06 | 0.222281 | 8 |
| E | 2.50 | 1.90 | 18.71 | 7.54 | 0.110964 | 6 |
| F | 0.89 | 0.24 | 2.35 | 0.95 | 0.432723 | 9 |
| G | -3.05 | 2.83 | 27.93 | 11.26 | 0.078519 | 5 |
| H | -7.48 | 17.02 | 167.83 | 67.65 | 0.014463* | 3 |
| I | 3.59 | 3.91 | 38.57 | 15.55 | 0.058715 | 4 |

**Table 4. Scheme and results of steepest ascent test.**

| No. | B | C | H | Slump error /% |
|-----|------|-------|-----|----------------|
| 1 | 0.1 | 0.1 | 0.1 | 49.88 |
| 2 | 0.95 | 0.325 | 0.3 | 19.53 |
| 3 | 1.8 | 0.55 | 0.5 | 8.36 |
| 4 | 2.65 | 0.775 | 0.7 | 4.48 |
| 5 | 3.5 | 1 | 0.9 | 8.21 |

can be applied to predict soil slump in paddy fields. The regression model for slump error is calculated as:

$$\varphi = 131.04 - 40.02B - 53.19C - 128.83H + 8.78BC + 13.26BH$$
$$- 80.17CH + 3.76B^2 + 56.95C^2 + 101.58H^2 \tag{6}$$

To analyze the influence of the interaction between the factors on the test index, the interaction diagrams of shear modulus and surface energy, shear modulus and soil-iron plate static friction coefficient and surface energy, and soil-iron plate static friction coefficient were drawn (Fig 2). Change in B and C resulted in a large change in surface slope; the contour line was elliptical indicating a significant interaction between B and C (Fig 2A). The changes in B and H also resulted in a significant change in surface slope (Fig 2B). The elliptical contour line indicates a significant interaction between B and H. Finally, changes in C and H resulted in a significant change in surface slope (Fig 2C). The elliptical contour line indicates a significant interaction between C and H [24].

## 3.4 Parameter optimization and validation

To obtain the best combination of parameters affecting the test index, the regression model was optimized using the optimization function in Design-Expert with the range of values of the test factors as the boundary conditions and the slump error as the target. The best combination of parameters was obtained; the soil surface energy was 3.257 J/m², soil shear modulus was 0.709 Mpa, soil-iron plate static friction coefficient was 0.701, and slump error was 1.812%. The simulation verification test of the optimized results under the best combination of parameters showed that the average error between the simulated and actual values of the slump was 2.04%. The simulation and actual test results are shown in Fig 3. The findings indicate that the discrete element model of paddy loam soil under this parameter is valid and can provide basic data for simulation analysis in the paddy pulping process.

## 4 Discussion and conclusion

In this paper, a discrete element simulation model of the paddy loam soil after soaking was established based on the slump test. The parameters of the discrete element model of the

**Table 5. Factor and codes of the Box-Behnken test.**

| Codes | Factors | | |
|-------|---------|---|---|
| | B | C | H |
| -1 | 1.8 | 0.55 | 0.5 |
| 0 | 2.65 | 0.775 | 0.7 |
| 1 | 3.5 | 1 | 0.9 |

**Table 6. Scheme and result of the Box-Behnken test.**

| No. | b | c | h | Slump error/% |
|-----|-----|-----|-----|-----|
| 1 | -1 | -1 | 0 | 14.00 |
| 2 | 1 | -1 | 0 | 3.18 |
| 3 | -1 | 1 | 0 | 10.89 |
| 4 | 1 | 1 | 0 | 6.79 |
| 5 | -1 | 0 | -1 | 17.91 |
| 6 | 1 | 0 | -1 | 7.23 |
| 7 | -1 | 0 | 1 | 8.05 |
| 8 | 1 | 0 | 1 | 6.39 |
| 9 | 0 | -1 | -1 | 8.33 |
| 10 | 0 | 1 | -1 | 17.32 |
| 11 | 0 | -1 | 1 | 10.02 |
| 12 | 0 | 1 | 1 | 4.58 |
| 13 | 0 | 0 | 0 | 3.14 |
| 14 | 0 | 0 | 0 | 2.62 |
| 15 | 0 | 0 | 0 | 3.89 |
| 16 | 0 | 0 | 0 | 2.49 |
| 17 | 0 | 0 | 0 | 3.44 |

paddy loam soil were calibrated using a combination of the actual test and discrete element simulation test with the slump error as the test index.

The basic parameters of the paddy loam soil after 24 h of soaking were determined. The average value of soil density was 1808 kg/m$^3$, the average water content was 32%, and the average slump was 177.4 mm.

The significant factors affecting the test indices were identified using the Plackett-Burman test. The optimum parameter interval was obtained from the steepest ascent test. The

**Table 7. Variance analysis of regression model.**

| Source | Sum of square | Degree of freedom | Variance | F | P |
|--------|---------------|-------------------|----------|-----|-----|
| Model | 388.52 | 9 | 43.17 | 90.13 | < 0.0001** |
| B | 92.89 | 1 | 92.89 | 193.94 | < 0.0001** |
| C | 2.05 | 1 | 2.05 | 4.28 | 0.0773 |
| H | 59.13 | 1 | 59.13 | 123.46 | < 0.0001** |
| BC | 11.29 | 1 | 11.29 | 23.57 | 0.0018** |
| BH | 20.34 | 1 | 20.34 | 42.47 | 0.0003** |
| CH | 52.06 | 1 | 52.06 | 108.69 | < 0.0001** |
| B$^2$ | 31.05 | 1 | 31.05 | 64.84 | < 0.0001** |
| C$^2$ | 35.00 | 1 | 35.00 | 73.08 | < 0.0001** |
| H$^2$ | 69.52 | 1 | 69.52 | 145.14 | < 0.0001** |
| Residual | 3.35 | 7 | 0.48 | | |
| Lack of Fit | 2.01 | 3 | 0.67 | 2.00 | 0.2569 |
| Pure Error | 1.34 | 4 | 0.34 | | |
| Cor Total | 391.87 | 16 | | | |

Note

** represents extremely significant influence (P < 0.01)

* represents significant influence (0.01 < P < 0.05).

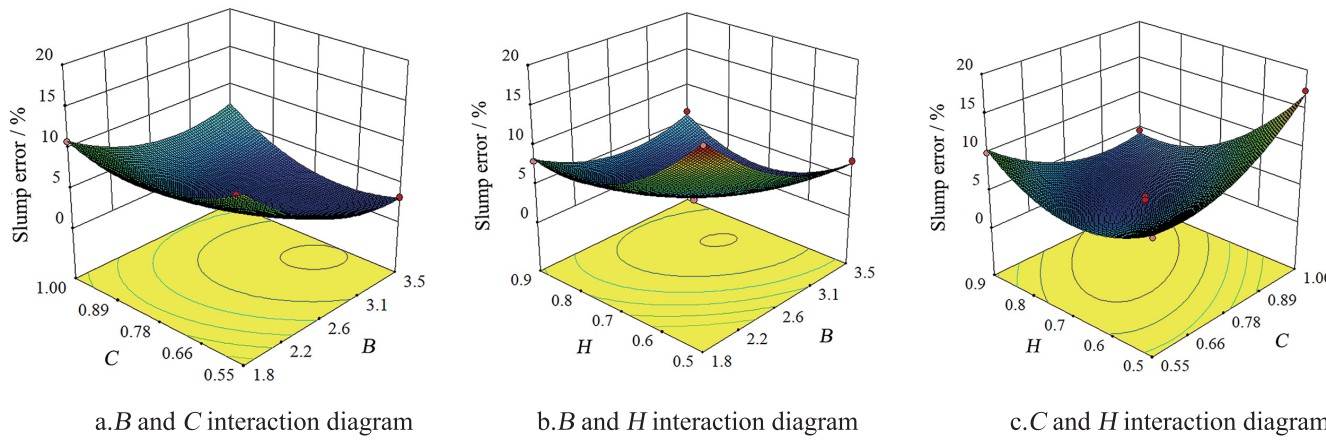

a.*B* and *C* interaction diagram          b.*B* and *H* interaction diagram          c.*C* and *H* interaction diagram

**Fig 2. Influence of interactions of factors on slump error.**

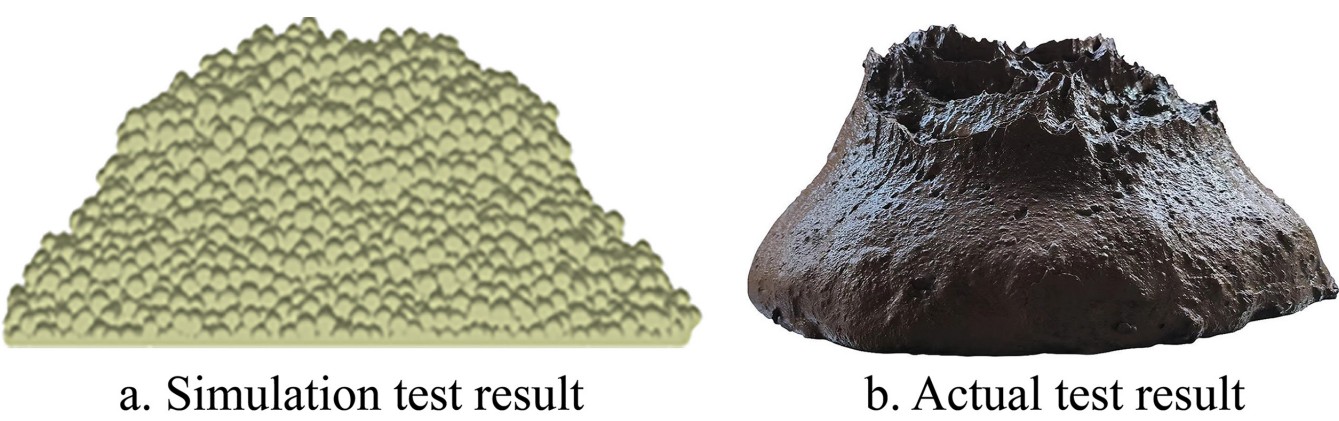

a. Simulation test result                                        b. Actual test result

**Fig 3. Result of simulation test and actual test.**

regression model of significant factors and test indices was obtained by the Box-Behnken test. The regression model was optimized to obtain the optimum parameters of the significant factors. The findings were surface energy 3.257J/m$^2$, soil shear modulus 0.709 Mpa, and static friction coefficient of soil-iron plate 0.701. Simulation verification using this combination of parameters yielded a slump error value of 2.04% for the paddy loam soil.

The collective results show that the developed discrete element model of paddy loam soil is accurate and can provide an accurate discrete element simulation model of paddy loam soil for the discrete element simulation process in the development of high-performance paddy pulping machines. However, since there are many types of rice growing soils, further studies on different rice growing soils will be needed to provide more comprehensive discrete element models of paddy soil for the development of pulping machines in different regions and to accelerate the application of the discrete element method in the field of paddy pulping.

## Supporting information

**S1 Fig. EDEM simulation test process.**
(TIF)

**S1 File.**
(DOCX)

## Author Contributions

**Data curation:** Hao Zhou.

**Funding acquisition:** Hao Zhou.

**Investigation:** Hao Zhou.

**Methodology:** Zhou Tienan, Hao Zhou, Jiangtao Ji.

**Project administration:** Zhou Tienan, Hao Zhou, Jiangtao Ji.

**Resources:** Zhou Tienan, Hao Zhou, Jiangtao Ji, Fengyun Sun, Zhiyu Qin.

**Software:** Hao Zhou, Jiangtao Ji, Fengyun Sun, Zhiyu Qin.

**Supervision:** Zhou Tienan, Hao Zhou.

**Validation:** Hao Zhou, Jiangtao Ji, Fengyun Sun.

**Visualization:** Zhou Tienan, Hao Zhou.

**Writing – original draft:** Zhou Tienan.

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
