## [Decision Letter · Decision Letter 0]

30 Mar 2023

PONE-D-23-07157Parameter calibration of discrete element simulation model for soaking field soil based on slump testPLOS ONE

Dear Dr. Zhou,

Thank you for submitting your manuscript to PLOS ONE. After careful consideration, we feel that it has merit but does not fully meet PLOS ONE’s publication criteria as it currently stands. Therefore, we invite you to submit a revised version of the manuscript that addresses the points raised during the review process.

We look forward to receiving your revised manuscript.

Kind regards,

Jiaolong Ren

Academic Editor

PLOS ONE

Journal Requirements:

2. In your Methods section, please provide additional information regarding the permits you obtained for the work. Please ensure you have included the full name of the authority that approved the field site access and, if no permits were required, a brief statement explaining why

Reviewers' comments:

Reviewer's Responses to Questions

**Comments to the Author**

1. Is the manuscript technically sound, and do the data support the conclusions?

Reviewer #1: Partly

Reviewer #2: Yes

2. Has the statistical analysis been performed appropriately and rigorously? 

Reviewer #1: Yes

Reviewer #2: Yes

3. Have the authors made all data underlying the findings in their manuscript fully available?

Reviewer #1: Yes

Reviewer #2: Yes

4. Is the manuscript presented in an intelligible fashion and written in standard English?

Reviewer #1: Yes

Reviewer #2: Yes

5. Review Comments to the Author

Reviewer #1: The manuscript needs further great improvement. There is currently no discussion section in the manuscript, and it is suggested that additional discussion be added to further explain the experimental results.

1. Introduction: The introduction section needs further improvement. For example, the author lists the results of six references, but does not make a comprehensive summary associated with the results. It's not clear. Authors need to give a discussion and evaluation associated with the research progress on development of high-performance paddy field pulping machine and the interaction mechanism between the implements and the soil. That is, what problems have been solved by existing research and what problems have not been solved? Which specific problems should be solved in this paper?

In the manuscript, readers may not be clear about the difference between this study and other researches, that is, where is the innovation?

2. In section for “2.1 determination of basic physical parameters”, if only one kind of soil is taken, this study is of little significance. It is necessary to add investigation to consider the effects of soil type or soil structure.

3. The authors should add some introduction to the methods in the manuscript, such as Plackett-Burman test, the steepest climb test, the Box-Behnken test.

4. Some of the experimental schemes in the manuscript should be explained in more detail:

(1) How to determine the value range of 9 parameters to be calibrated according to the literature?

(2) How is the combination of parameter levels involved in the experimental protocol determined in table 2

(3) Are the values of the 4 levels and their corresponding parameters fixed in table 4?

(4) According to Table 5, there are 27 schemes can be formed, why only 17 schemes are selected in the study, and the values of parameters b,c and h are the same for schemes after No.14

5. The slump error of paddy soil is used as a test index to calibrate the discrete element simulation parameters in the manuscript.

The guiding significance for the study of interaction mechanism between the implements and the soil, or for t development of high-performance paddy field pulping machine is not clear.

Reviewer #2: There were several typos, grammatical errors, and incomplete sentences. Authors are advised to edit the manuscript for better clarity.

Add better and high quality images.

Recent literature is sufficiently cited. Please check all the references one by one to avoid any mistakes in-text citations.

6. PLOS authors have the option to publish the peer review history of their article (what does this mean?). If published, this will include your full peer review and any attached files.

Reviewer #1: No

Reviewer #2: No

---

## [Author Response · Author response to Decision Letter 0]

16 Apr 2023

Reviewer 1: I have incorporated all of your suggestions into my revision. They were very helpful. Thank you. 

Reviewer 2: I have incorporated all of your suggestions into my revision. Thank you for your help.

---

## [Decision Letter · Decision Letter 1]

24 Apr 2023

Parameter calibration of discrete element simulation model for soaking paddy loam soil based on slump test

PONE-D-23-07157R1

Dear Dr. Zhou,

We’re pleased to inform you that your manuscript has been judged scientifically suitable for publication and will be formally accepted for publication once it meets all outstanding technical requirements.

Kind regards,

Jiaolong Ren

Academic Editor

PLOS ONE

Additional Editor Comments (optional):

Reviewers' comments:

Reviewer's Responses to Questions

**Comments to the Author**

1. If the authors have adequately addressed your comments raised in a previous round of review and you feel that this manuscript is now acceptable for publication, you may indicate that here to bypass the “Comments to the Author” section, enter your conflict of interest statement in the “Confidential to Editor” section, and submit your "Accept" recommendation.

Reviewer #1: All comments have been addressed

2. Is the manuscript technically sound, and do the data support the conclusions?

Reviewer #1: Yes

3. Has the statistical analysis been performed appropriately and rigorously? 

Reviewer #1: Yes

4. Have the authors made all data underlying the findings in their manuscript fully available?

Reviewer #1: Yes

5. Is the manuscript presented in an intelligible fashion and written in standard English?

Reviewer #1: Yes

6. Review Comments to the Author

Reviewer #1: The findings of this paper are valuable in optimizing the parameters of the paddy beating device to improve its working quality and efficiency. I suggest to accept.

7. PLOS authors have the option to publish the peer review history of their article (what does this mean?). If published, this will include your full peer review and any attached files.

Reviewer #1: No

---

## [Editor Report · Acceptance letter]

22 May 2023

PONE-D-23-07157R1 

Parameter calibration of the discrete element simulation model for soaking paddy loam soil based on the slump test 

Dear Dr. tienan:

I'm pleased to inform you that your manuscript has been deemed suitable for publication in PLOS ONE. Congratulations! Your manuscript is now with our production department. 

Kind regards, 

on behalf of

Dr. Jiaolong Ren 

Academic Editor

PLOS ONE